# The Basic Properties of Gold Nanoparticles and their Applications in Tumor Diagnosis and Treatment

**DOI:** 10.3390/ijms21072480

**Published:** 2020-04-03

**Authors:** Xue Bai, Yueying Wang, Zhiyun Song, Yanmin Feng, Yuanyuan Chen, Deyuan Zhang, Lin Feng

**Affiliations:** 1School of Mechanical Engineering & Automation, Beihang University, Beijing 100191, China; xuebai@buaa.edu.cn (X.B.); wyying1117@gmail.com (Y.W.); szy469350797@buaa.edu.cn (Z.S.); fengyanmin@buaa.edu.cn (Y.F.); chenyuanyuan0526@sina.com (Y.C.); zhangdy@buaa.edu.cn (D.Z.); 2Beijing Advanced Innovation Center for Biomedical Engineering, Beihang University, Beijing 100083, China

**Keywords:** gold nanoparticle, tumor diagnosis and treatment, cancer nanotechnology, drug delivery

## Abstract

Gold nanoparticles (AuNPs) have been widely studied and applied in the field of tumor diagnosis and treatment because of their special fundamental properties. In order to make AuNPs more suitable for tumor diagnosis and treatment, their natural properties and the interrelationships between these properties should be systematically and profoundly understood. The natural properties of AuNPs were discussed from two aspects: physical and chemical. Among the physical properties of AuNPs, localized surface plasmon resonance (LSPR), radioactivity and high X-ray absorption coefficient are widely used in the diagnosis and treatment of tumors. As an advantage over many other nanoparticles in chemicals, AuNPs can form stable chemical bonds with S-and N-containing groups. This allows AuNPs to attach to a wide variety of organic ligands or polymers with a specific function. These surface modifications endow AuNPs with outstanding biocompatibility, targeting and drug delivery capabilities. In this review, we systematically summarized the physicochemical properties of AuNPs and their intrinsic relationships. Then the latest research advancements and the developments of basic research and clinical trials using these properties are summarized. Further, the difficulties to be overcome and possible solutions in the process from basic laboratory research to clinical application are discussed. Finally, the possibility of applying the results to clinical trials was estimated. We hope to provide a reference for peer researchers to better utilize the excellent physicochemical properties of gold nanoparticles in oncotherapy.

## 1. Introduction

The original application of gold nanoparticles (AuNPs) in the biomedical field can be traced back to the Middle Ages, the so-called potable gold, known in old Latin texts as *aurum potabile*. Potable gold behaved as a fabulous curative power for various diseases, such as heart and venereal problems, arthritis, epilepsy and tumors, as well as for diagnosis of syphilis [1]. However, due to the lack of understanding the nature of AuNPs, the colloidal gold used is mostly in the oxidation state, so there are many serious side effects in the use process. Thanks to the development of modern nanotechnology, the biomedical application of AuNPs have made tremendous progress and continues to draw considerable research attention. In view of this, many previous reviews have summarized the superior properties of AuNPs and their applications in the field of biomedicine from different perspectives. Ghosh and Pal expound the interparticle coupling effect on the surface plasmon resonance (SPR) of AuNPs through specialized explanations of assembling strategies, optical properties and applications [2]. Distinct from other published reviews, this review focuses more on the basic theories of the SPR of AuNPs, which requires readers to have certain expertise in this area. Dreaden et al. described the synthesis and functionalization of a wide variety of AuNPs and their applications in diagnostics, imaging and medicine [3]. What distinguishes this review from others is that it describes the metabolic dynamics of AuNPs in vivo. Dykman and Khlebtsov published a review focusing on the biomedical applications of AuNPs, particularly their immunological properties [4]. Yang et al. introduced the chemical synthesis, optical properties, biomedical applications and the pharmacokinetics of AuNPs. In contrast to other reviews, one focused on the tunable optical properties of AuNPs [5]. Other specific references on the physical and chemical properties and applications of AuNPs are listed in Appendix A. Through these reviews, it is not difficult to find that the physical and chemical properties of AuNPs are greatly affected by their nanostructure (shape and crystal texture). This effect will eventually affect their biomedical applications. For example, compared with spherical AuNPs, gold nanorods or gold nanostars are more suitable for photothermal therapy (PTT) or photodynamic therapy (PDT), because they can absorb near-infrared light (NIR) more efficiently [6]. Au clusters can produce propylene oxide (PO) by using only O_2_ and water, whereas Au NPs cannot [7]. This will be covered in later paragraphs.

In a series of biomedical applications of AuNPs, its role in tumor diagnosis and treatment is particularly prominent. In order to make AuNPs more suitable for tumor diagnosis and treatment, their natural properties should be systematically and profoundly understood. In this review, we focused on disclosing the fundamental properties of AuNPs and the interrelationships between these properties. The fundamental properties of AuNPs were discussed from two aspects: physical and chemical. After the properties are introduced, their applications for tumor diagnosis and treatment are also listed. These applications include image agent, phototherapy, radiotherapy, targeting, nano-enzyme and drug delivery (Figure 1). The clinical trials of AuNPs with various physical and chemical properties are listed to provide a reference for the clinical transformation of the scientific research achievements.

## 2. Physical Properties 

Among the physical properties of AuNPs, localized surface plasmon resonance (LSPR), radioactivity and high X-ray absorption coefficient are widely used in the diagnosis and treatment of tumors. The LSPR of AuNPs can further lead to surface-enhanced Raman spectroscopy (SERS), surface enhanced fluorescence (SEF), photothermal conversion, photochemical conversion and colorimetric responses. These excellent qualities have been widely applied for non-invasive detection in vivo and in situ, imaging, PTT, PDT and in vitro diagnostics (IVD). The radioactivity of AuNPs can be used for radiotherapy and radionuclide imaging (RNI). The high atomic number of AuNPs has been explored for radiotherapy sensitization.

### 2.1. Localized Surface Plasmon Resonance (LSPR) 

Under the stimulation of light, conduction electrons on a noble metal oscillate collectively, which is called “plasmon” [8]. The plasmon resonance (PR) is an absorption band that occurs when the incident photon frequency is resonant with the collective oscillation of the conduction electrons [6]. The disturbance of the incident electromagnetic wave on the metal decreases rapidly with the depth of the metal, so the resonance often occurs on the metal surface, which is known as surface plasmon resonance (SPR) [9]. When the SPR is restricted to smaller volumes, so-called nanoparticles, which are comparable in size to the wavelength of the incident light, the LSPR occurs. With the help of LSPR, AuNPs present two important effects: an enhancement effect of the local electromagnetic field and an extinction coefficient (Figure 2).

When the LSPR occurs, the electromagnetic fields near the AuNP surface can be enhanced up to several orders of magnitude, and the largest enhancement occurs in the areas of highest local curvature (hot spots) [10]. The “hot spots” can enhance the spectral signals of the substances near their surface, which is called the surface enhancement spectrum (SES). SES studies related to oncotherapy mainly focus on SERS and SEF [11].

When the LSPR occurs, the optical extinction of the AuNPs can be maximized (e.g., 2.7 × 10^8^ M^–1^ cm^–1^ for 13 nm AuNPs), more than 1000 times stronger than ordinary organic molecules [12], which strongly enhanced the photothermal conversion efficiency, photochemistry conversion and light energy absorption of the AuNPs. These properties can be used for the PTT, PDT and colorimetric assays in tumor diagnosis and in treatment facilities. 

In this section, we focus on the physical properties of AuNPs that depend on LSPR and the relationship between these physical properties and LSPR. The size, shape, composition and microenvironment can significantly affect the LSPR of AuNPs. Readers interested in this area can consult the excellent reviews cited in Reference [13].

#### 2.1.1. Surface-Enhanced Raman Spectroscopy

When monochromatic light reaches the surface of a particle, Rayleigh scattering occurs when the particle is much smaller than the wavelength of light. Theoretically, the frequency and direction of the scattered light are the same as that of the incident light, but the molecules adsorbed on the surface of the particles also vibrate when receiving the incident light, resulting in the deviation of the frequency of part of the scattered light and the original incident light (Stokes scattering and anti-Stokes scattering). The energy difference between the scattered photon and the incident photon is called Raman scattering (RS) [14], which corresponds to the energy of the vibrational mode of the adsorbed molecule. Therefore, a Raman image can provide highly resolved vibrational information (≈0.1 nm), which is called a chemical fingerprint, to ensure the accuracy of single molecule detection [15]. However, the cross sections of RS are about 10–30 cm^2^/molecule, which make the signal sensitivity of RS too low to be detected [16]. This serious limitation of RS can be overcome using SERS, which can enhance the Raman signal from 10^6^ to 10^15^ and therefore is widely used in sensing, detection and imaging applications for cancer diagnosis [17]. AuNPs, as contrast agents of SERS, are used to image the smaller tumors, distinguish tumor cells, monitor tumor metabolism and detect tumor markers. 

Qian and co-workers used AuNPs conjugated with single-chain variable fragment (ScFv) antibodies to target epidermal growth factor receptors (EGFR) on human cancer cells and in xenograft tumor models. The results have shown that the RS signal of these nanoparticles can be enhanced and detected [18]. Hossain and co-workers used a homogenous AuNP-deposited ITO substrate to enhance the Raman signals and successfully characterized and distinguished two different sub-types of breast cancer cells that originated from the same or different organs [19]. Shiota and co-workers used AuNPs with horse-bean shapes, which can generate many SERS excitation sources, to visualize the anti-oxidant consumed hypotaurine in the global metabolic reprogramming of cancer. This is an important mechanism in cancer survival [20]. Lin and co-workers used AuNP-based SERS to detect the serum from patients (*n* = 38) and volunteers (*n* = 45). The results showed that the diagnostic sensitivity and specificity of the colorectal tumor markers can reach 97.4% [21]. For more information on the application of AuNPs as SERS imaging agents, please refer to other literature reviews (e.g., [22]).

#### 2.1.2. Surface Enhanced Fluorescence

The LSPR of AuNPs can produce two completely opposite effects on the signal strength of fluorescent molecules: fluorescence quenching and fluorescence enhancement. This depends on the distance between the fluorescence molecules and the AuNPs. The free fluorescent molecules will jump from the ground state to an excited state after absorbing photons. The excited fluorophores will fall back to the first excited state after vibrational relaxation with a rate of internal conversion (*k*_int_) and then fall back to the ground state with energy decay. The total energy decay rate (Γ) consists of the radiative decay rate (γ) and nonradiative decay rate (*k*_nr_). Radiative decay appears as fluorescence emission [23].

The nonradiative decay rate appears as intersystem transitions, energy transfer, thermal radiation and so on. In these non-radiative decay forms, fluorescence resonance energy transfer (FRET) is the primary reason for the fluorescence quenching of AuNPs. The FRET between fluorescence molecules and AuNPs shows a marked decrease with distance. When the distance is less than 5 nm, the energy of the excited fluorescent group will be transferred to the AuNPs in totality, resulting in fluorescence quenching [24]. The quenching and coming back of the fluorescence can represent the interaction among the particles so that they can be used for diagnostics. The common practice is to link AuNPs and fluorescent molecules with different linkers. The linker interacts with the substance to change its structure, which further changes the distance between the AuNPs and fluorescent molecules. For example, researchers select the recognition sequences as the linker to recognize cancer cells in lung tumors [25], acute lymphocytic leukemia [26] and human Burkitt’s lymphoma [27]. This method is also used to determine the detection of intracellular mRNA [28] and even the simultaneous detection of multiple mRNA [29] in living cancer cells. 

The enhancement effect of LSPR on SEF can be attributed to two factors: enhance the excitation efficiency and decay rate of fluorescence. First, the enhancement electromagnetic fields of “hot spots” lead to enhanced excitation of the molecules placed there. Second, AuNPs act as a resonant cavity to enhance the radiative rate of a quantum emitter, due to the increased local density of states [30]. Thanks to this excellent property, AuNPs are widely used as two-photon luminescence (TPL) imaging agents to perform tumor cell imaging [31] and three-dimensional in vivo imaging [32]. In addition, AuNPs can be used to connect quantum dots, fluorescent molecules or even the autofluorescence emission intensity from the cell, to improve the detectability of tumor cells and guided tumor therapy [33]. 

#### 2.1.3. Photothermal Conversion

After absorbing photons, AuNPs convert the light energy of the electrons into kinetic energy. The moving electrons are scattered by the lattice/phonon, and part of the kinetic energy is transformed into vibration energy of the lattice. The vibration energy obtained by the lattice is finally expressed in the form of heat. This is the so-called “photothermal effect”. Compared with normal tissue blood vessels, tumor vascular variety has a lot of holes in the pipe wall, leading to tumor tissue inside the blood supply clogging; so, the tumor tissue cooling ability is worse than normal tissue, so the heat produced in the process of heat energy is more likely to accumulate within the tumor and its temperature can easily achieve 46 °C above, while the surrounding normal tissues can only be heated to 41 °C or so. In addition, tumor cells are inherently less heat-resistant than normal cells, with the former usually having a lethal temperature of 42.5~43 °C, while the latter can withstand temperatures as high as 47 °C. In addition, tumor cells are inherently less heat-resistant than normal cells, with the former usually having a lethal temperature of 42.5~43 °C, while the latter can withstand temperatures as high as 47 °C. Therefore, photothermal conversion has a promising prospect in cancer cell imaging and PTT [34]. Biological tissue has very low extinction and spontaneous fluorescence in the NIR [35], which is the most suitable excitation light for PTT. The surface plasmon absorption maximum of the AuNPs can be adjusted to the NIR region to achieve efficient photothermal conversion, which is called plasmonic-PTT (PPTT) [36]. In 2003, Lin and co-workers first demonstrated AuNPs have a photothermal effect after exposure to visible light in vitro [37]. Then, the PTT application of AuNPs under NIR in vivo was demonstrated through being interstitially injected [38] and intravenously injected [39]. After this, many methods have been used to modify the LSPR of AuNPs to improve the efficiency of the photothermal conversion and to be more suitable for in vivo application [40]. These methods include inducing the aggregation of small AuNPs (<10nm) and change the nanostructure of the AuNPs. For more information on the application of AuNPs in photothermal therapy, please refer to [41].

The photothermal conversion characteristics of AuNPs can be supervised by photothermal imaging (PTI) [34] and photoacoustic imaging (PAI), which can be effectively applied in clinical diagnosis of tumors, assisting or guiding the hyperthermia process as well as realizing the integration of visual therapy and diagnosis and treatment [42]. For more information on the application and principle of AuNPs in PTT and PAI, please refer to [43].

#### 2.1.4. Photosensitization

Upon photoexcitation of the AuNPs, the excited state photon energy can undergo energy transfer to neighboring molecules, such as the organic photosensitizer or molecular oxygen (O_2_). This will lead to the generation of cytotoxic oxygen-based species (singlet oxygen (^1^O_2_), O^2−^ and OH^−^, etc.), which is known to play a central role in the PDT treatment of cancer [44]. There are basically two major aspects concerning AuNP-based PDT, namely (1) using AuNPs to promote the photosensitization of photosensitizers to produce ^1^O_2_ and (2) AuNPs alone can sensitize the formation of ^1^O_2_. Thanks to the LSPR, AuNPs can efficiently absorb the energy of NIR and transfer it to a photosensitizer or O_2_, which resolves some drawbacks of common organic photosensitizers, such as poor photo-stabilities, being easily degraded by enzymes and a low efficiency of light energy conversion [45]. As a photosensitive enhancer, AuNPs can transfer absorbed light energy to the conjugated organic photosensitizer and promote its sensitization to generate ^1^O_2_ [46]. As a photosensitizer, AuNPs can transfer absorbed light energy directly to O_2_ to produce ^1^O_2_ and kill tumor cells [47]. In 2011, Vankayala and co-workers originally demonstrated that ^1^O_2_ can be formed through direct sensitization by AuNPs under visible light exposure [47]; they were the first to report that this sensitization can be implemented under NIR (915 nm) light excitation and can exert dramatic PDT effects on the destruction of solid tumors in mice [48], and that the yield of ^1^O_2_ is related to its shape [49]. At present, people are actively exploring to further improve the role of AuNPs in photodynamic therapy. For example, Chen and co-worker developed an AuNP-based photosensitizing agent that simultaneously enables fluorescence imaging, tumor hypoxia relief and NIR-II (1064 nm) light-induced PDT in vivo [50].

In many studies, PTT and PDT are often used in combination. These applications are listed in Table 1.

#### 2.1.5. Colorimetric Responses

Thanks to the high molar absorption coefficient, the detection sensitivity of AuNPs by colorimetric analysis reach to nanomole level, which is much lower than the traditional colorimetric method. AuNP-based colorimetric biosensing assays take advantage of the color change of the AuNP-induced plasmon from red to blue, purple or gray when the AuNPs were turned to aggregate the analytes. These analytes include tumor-related proteins, nucleic acids or cytokines [62]. In 2010, Kang and co-workers first applied AuNPs-based colorimetric assay for cancer diagnosis by charge-induced AuNP aggregation [63]. In this study, they selected activated protein kinase Cα (PKCα) as the cancer marker [64], cationic PKCα-specific peptide as the substrate and AuNPs with anionic surface charges as the chromogenic agent. When the system was applied to cancer cells or tissue lysates, the cationic peptide substrate was phosphorylated and increased the anionic charges in the phosphorylation site, which suppressed AuNP aggregation, resulting in a red color. On the application of normal cells or tissue lysates, a blue color is produced. At the same time, Lu and co-workers used AuNP-based colorimetric assays for highly selective and sensitive detection of breast cancer with high HER2 expression from other breast cancer cells by antibody-induced AuNP aggregation [65]. In 2011, Lee and co-workers used AuNP-based colorimetric assays for ultra-selective detection of cancer mutation points (BRCA1) by base pairing-induced AuNP aggregation [66]. This system is also used for detection of the mutation of the epidermal growth factor receptor gene (EGFR) [67], the Kirsten rat sarcoma viral oncogene homologue gene (KRAS) [68] or p53 [69]. In the AuNP-based colorimetric assay, the signal can be amplified through isothermal [68] or polymerase chain reaction [70]. This allows the cancer-related molecule with picomolar sensitivity to be visualized, which is suitable for point-of-care applications.

### 2.2. Radioactivity

As one of the metals, Au is a radionuclide, having nuclear properties. Thanks to the desirable nuclear properties, ^198^Au (t_1/2_ = 2.7 d) and ^199^Au (t_1/2_ = 3.2 d) can be excreted intact in the urine [71] or liver [72], and are generally used for biomedicine. ^198^Au can emit β-particles with a maximum energy of 0.96 MeV and a 412 keV γ-ray [73]. ^198^AuNPs contain a high percentage of these radioactive atoms and therefore require the use of fewer amounts of NPs to achieve the desired amount of radioactivity for treatment and imaging [74].

The high energy of β-emission ^198^AuNPs is effective for the destruction of tumor cells/tissue [75]. For example, in the mouse model of human prostate, the tumors received a 70 Gy dose when the mice were intravenously injected with ^198^AuNPs and significantly inhibited tumor volume [76]. Target-specific ^198^AuNPs was developed to reduce the injected dose of ^198^AuNPs to get the similar therapeutic effect. The tumor-specific antibodies or the nanocomposite device was used to increase the accumulation in tumor tissue.

The γ-rays can readily penetrate soft tissues free from interferences and are suboptimal for single-photon emission computed tomography (SPECT) scanners [77], therefore AuNP-based drug carriers can be imaged in vivo [78]. Chen and co-workers produced a PEGylated ^198^AuNP and applied it to image and trace AuNPs in living animals and dissected tissues [79]. In addition to SPECT imaging, ^198^AuNPs were also synthesized for Cerenkov luminescence imaging [80].

^199^Au can emit β-particles with a maximum energy of 0.45 MeV and a 208 keV γ-ray, which is more suitable for SPECT image than ^198^Au [81]. For example, a ^199^AuNPs was synthesized by doping ^199^Au directly into the nanocrystals and performed SPECT tumor imaging in a mouse triple-negative breast cancer (TNBC) model in a clinical study [82]. Fazaelia and co-worker synthesized the amino-functionalized graphene oxide sheets labeled by ^198,199^Au NPs for SPECT imaging in rats bearing fibrosarcoma tumors [83]. The initial interest of ^199^Au was from its ability to attach to monoclonal antibodies selectively. This allows ^199^AuNPs to be modified with more tumor-targeted monoclonal antibodies and could result in a higher specific activity, subsequently increasing the delivery of therapeutic payloads to the tumors [84]. For more information on the role of AuNPs in tumor radiographic imaging and treatment, please refer to the relevant reviews (e.g., [85]).

### 2.3. High Atomic Number

When the atomic number is higher than 53, the absorbed dose of the X-ray will increase [86]. Au has an atomic number of 79, which strongly increases the X-ray absorption coefficient [87]. This can minimize the damage of X-ray to normal tissues and becomes a promising radiotherapy sensitizer for oncotherapy [88]. Au increases the cross section of the cancer cell or tissue reaction to X-rays, and emit photoelectrons [89], Auger electrons [90], Compton electrons [91] and other secondary electrons [92]. These secondary electrons can ionize DNA molecules directly, which further break the DNA strand, base and sugar crosslinking [93]. What is more important, these secondary electrons can react with water in tissues to generate free radicals, which bind to DNA, causing electron transfer of target molecule DNA and oxidation [94]. This is the physical explanation of the mechanism of radiotherapy sensitization for AuNPs. This X-ray enhancement effect is usually expressed as a dose enhancement factor (DEF) [95]. The DEF of AuNPs was calculated by the theoretical model under different conditions, such as the diameter and concentration of the nanoparticles [96], X-rays of different energies [97], intracellular localization [98] and three-dimensional dose distribution [99]. The radiosensitization effect of AuNPs was initially highlighted based on the experiment of Hainfeld and co-workers [100]. Radiotherapy with AuNPs at the same dose significantly increased the survival time of tumor mice. 

In addition of radiotherapy sensitization, the high absorption rate of AuNPs to X-rays makes it possible to be used as an X-ray (CT) imaging agent for cancer diagnosis, tumor therapy guiding or therapeutic evaluation. In 2008, Popovtzer and co-workers first presented in vitro proof of the molecular CT imaging of cancer based on the target AuNPs [101] and then demonstrated in vivo the feasibility of cancer diagnosis at the cellular and molecular level with a standard clinical CT [102]. They selected anti-EGFR to modify the AuNPs and injected them into human squamous cell carcinoma head and neck cancer xenograft mice through the tail vein. Small tumor tissues ingesting these target AuNPs can be clearly seen through CT. AuNP-based CT imaging was then developed for guiding cancer therapy through radiation or photo-thermal therapy. For a more detailed theory and application of gold nanoparticles in tumor radiotherapy, please refer to [95]. The author introduced the mechanisms of AuNP radiosensitization from three aspects: physics, chemistry and biology.

## 3. Chemical Properties

### 3.1. Easy to Couple

As an advantage over many other nanoparticles, AuNPs can form stable chemical bonds with S-and N-containing groups. This allows AuNPs to attach to a wide variety of organic ligands or polymers with a specific function. These surface modifications endow AuNPs with outstanding biocompatibility, targeting and drug delivery capabilities [103].

#### 3.1.1. Biocompatibility

The biocompatibility of AuNPs is through their biological fate in vivo, which can be evaluated by pharmacokinetics, tissue distribution, toxicity and clearance. Biocompatibility is the essential condition for all application of AuNPs in vivo, which can be improved by surface modification, most of which is based on the formation of Au–S bonds. Measures to optimize the pharmacokinetics of AuNPs is to increase the circulation half-life through reducing the clearance by the mononuclear phagocyte system (MPS) or optimize the physical size. Polyethylene glycol (PEG) has been widely applied to decrease phagocytosis of AuNPs by MPS and the circulation half-life increases as the length of the PEG chains increases [104]. Compared to 100 nm AuNPs, 15 nm AuNPs can circulate in vivo for a longer time [105]. However, smaller AuNPs (less than 6 nm) will be rapidly filtered out and cleared by the kidneys [106]. AuNPs used for tumor diagnosis and treatment need to increase their retention in tumor tissues and reduce their accumulation in other tissues. AuNPs enter the blood circulation and can reach opsonization within minutes, which leads to the formation of a protein crown on the AuNP surface to facilitate the recognition by phagocytic cells in the MPS. The MPS involves the liver, spleen and bone marrow. Therefore, most AuNPs are removed from the bloodstream and primarily up-concentrated in the liver, spleen and bone marrow [107]. Compared to a neutral surface, a charged surface of the AuNPs are more likely to form protein crowns [40]. AuNPs deposited in other organs outside the tumor tissue can cause toxicity, mainly manifested as an acute inflammatory response and cell apoptosis. For example, 13 nm AuNPs can disturb the metabolism of low-density lipoprotein in cells [108]; 8 to 37 nm AuNPs induced fatigue, loss of appetite, change of fur color and weight loss in mice, even death [109]. Fortunately, these toxic effects can be addressed by surface modification and physicochemical parameter optimization. To mitigate the toxic side effects, AuNPs must be quickly cleared from the healthy organs. The clearance of AuNPs from the body depend on the renal and biliary clearance, because AuNPs cannot be digested by enzymes within the body. Small AuNPs (less than 10 nm) are more easily removed than large ones. Small AuNPs can be removed from the body by renal clearance, up to 70% of it within 72 h [110]. Even if the detailed mechanism is unclear, the studies have shown that small AuNPs can be removed from the body via bile [111]. Interested readers can obtain more information from other reviews (e.g., [5,6,112]).

#### 3.1.2. Targeting

The targeting of AuNPs can be introduced from two aspects: passive targeting (enhanced permeability and retention (EPR) effect and MPS escape) and active targeting (tumor cell targeting and stimuli-response). Due to the rapid growth of the tumor, the internal blood vessels are defective and the lymphatic vessels are underdeveloped [113]. This allows AuNPs of a certain size to pass through tumor vessels efficiently and accumulate in tumor sites, which has been applied for tumor imaging and therapy [114]. The MPS escape of AuNPs can be implemented through coatings of hydrophilic polymers [115], forming the particles from branched [116] or with hydrophilic and hydrophobic domains [117]. The tumor cell targeting of AuNPs mainly relies on the ligand and can be recognized by cancerous cells, such as Lam 67R [85] and GRP [118] human prostate tumors, and CCR5 and HER2 [65] for breast tumors. The stimuli-response includes exogenous stimuli, such as NIR [119], magnetic [120], ultrasound [121] and endogenous stimuli, such as Ph [56] and redox [122].

#### 3.1.3. Delivery

AuNPs can be combined with chemotherapy drugs, proteins or nucleic acids through electrostatic adsorption or covalent bonds. This property, together with its superior biocompatibility and targeting, makes it the most promising delivery for tumor targeting. For chemotherapy, AuNPs can carry mitoxantrone (MTX) [123], phthalocyanine4 (Pc4) [124], doxorubicin (Dox) [125] and photosensitizer [46] to improve their tumor targeting and enhance its therapeutic effect. For immunotherapy, AuNPs can be loaded with immune target antibodies and stimulate the activation of immune cells. These antibodies include polyclonal anti-carcino-embryonic antigen (CEA) [126], monoclonal anti-HER2 [121] and so on. Dumani et al. designed AuNPs coated with glycol-chitosan that can be taken up by immune cells and subsequently transported to the sentinel lymph node to detect metastases [121]. AuNPs can modified with nucleotide sequence for detecting [127], imaging [25] and therapy [26].

### 3.2. Catalytic Activity and Applications

In 1987, Haruta et al. reported that AuNPs can catalyze the oxidation of CO at or even below room temperature with extraordinary efficiency [128]. Based on this pioneering work, the study of AuNP-based biomimetic catalysts has also developed rapidly [129]. So far, the mimic enzyme activity of AuNPs has been reported that include nuclease, esterase, silicatein, glucose oxidase (GO), peroxidase (POD), catalase, superoxide dismutase and oxidase. In the mimic enzyme activities of the AuNPs, POD and GO mimic activity have been used in the diagnosis and therapy of tumors.

The POD-mimic activity of AuNPs was first discovered by catalyzing the oxidation of the peroxidase substrate 3,3,5,5-tetramethylbenzidine (TMB) by H_2_O_2_ to develop a blue color in an aqueous solution [130]. This has been used to catalyze H_2_O_2_ located in tumor cells to generate ·OH under an acidic pH for intracellular oxidative damage of gastric tumors [131] or to detect the concentration of GSH for cancer diagnosis [132].

The GO-mimic activity of AuNPs was first demonstrated through the aerobic oxidation of glucose to gluconate and H_2_O_2_, which is regarded as a model reaction [133]. The production of H_2_O_2_ in situ can induce AuNP growth that can detect cancer biomarkers at the picomolar levels [134]. The H_2_O_2_ also can be used as a substrate that is catalyzed by POD to liberate high-toxic hydroxyl radicals for inducing tumor-cell death [135].

### 3.3. Biological Activity

AuNPs not only play a role in tumor therapy through extrinsic functionalization or activation, but also have intrinsic antineoplastic biological activity. This property is often associated with the size of the AuNPs. The smaller AuNPs (less than 2 nm) can induce cellular oxidative stress, mitochondrial damage and DNA interactions to kill cancer cell. The larger AuNPs did not show the same killing effect at the same concentration [136]. AuNPs with a particle size of about 5 nm can selectively interact with heparin-binding glycoproteins on the surfaces of endothelial cells and subsequently inhibit tumor activity by changing the conformation of the molecules [137]. In addition, the AuNPs have also been reported to enhance the apoptosis and inhibit the proliferation of cancer cells [138]. AuNPs have also been demonstrated to selectively accumulate in the mitochondria of tumor cells, decrease mitochondrial electrical potential, increase reactive oxygen species and eventually lead to apoptosis of tumor cells, while normal cells and stem cells have not shown the same effect [139]. Interested readers can refer to [3,4].

## 4. Application of AuNPs in Clinical Trials

In practical applications, especially in clinical trials, it is often necessary to comprehensively apply the physical and chemical properties of AuNPs to achieve the purpose of oncotherapy. The current applications of AuNPs in clinical trials are listed in Table 2.

## 5. Challenges and Prospects

Compared with the large amount of exciting data obtained in laboratory studies, there are very few tumor therapy strategies based on AuNPs that are actually currently used in clinical trials or entering clinical trials. The reason is that these strategies face many challenges in the process of clinical transformation from laboratory studies, such as how to evade the clearance of the mononuclear phagocytic system (MPS) or renal excretion and thus reach tumor tissue efficiently; how to cross multiple physiological barriers and play a therapeutic effect; and how are nanoparticles metabolized out of the body after treatment. The most promising solution is to rationalize the application of the physical and chemical properties of AuNPs. For example, Liu’s research group designed a multifunctional poly (amino acid)–gold–magnetic complex that could escape renal clearance because of its big-particle-diameter (≈170 nm); can cross the vascular barrier through the EPR effect; be used for CT imaging based on the high X-ray absorption coefficient; efficient uptake due to the guanidine group in coupling the arginine of the AuNPs; perform PTT by taking advantage of the high photothermal conversion; and can be cleared renally due to biodegradation into small particle sizes (≈3 nm). On the basis of studying the excellent physical and chemical properties of AuNPs, more AuNP-based tumor therapy strategies will enter clinical trials and be approved for the treatment of tumor patients.

## Figures and Tables

**Figure 1 ijms-21-02480-f001:**
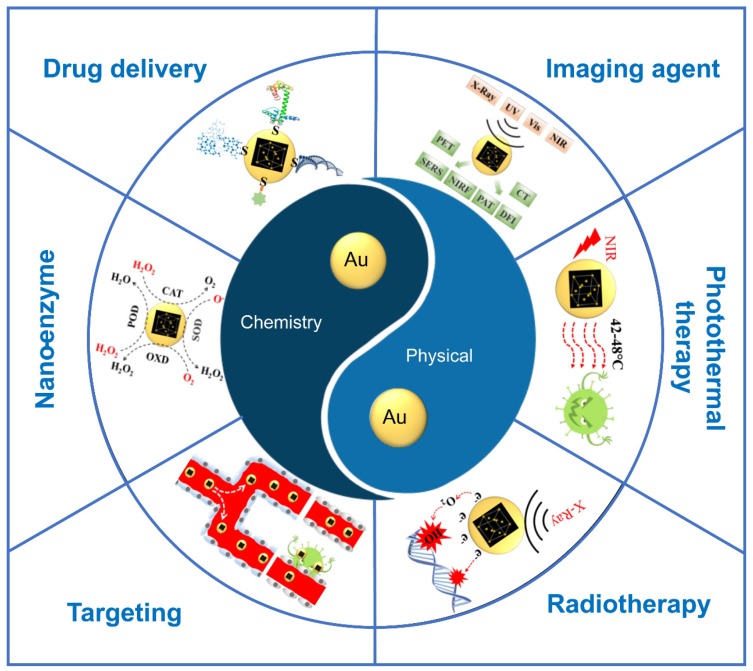
Applications for tumor diagnosis and treatment based on the basic physical and chemical properties of gold nanoparticles (AuNPs).

**Figure 2 ijms-21-02480-f002:**
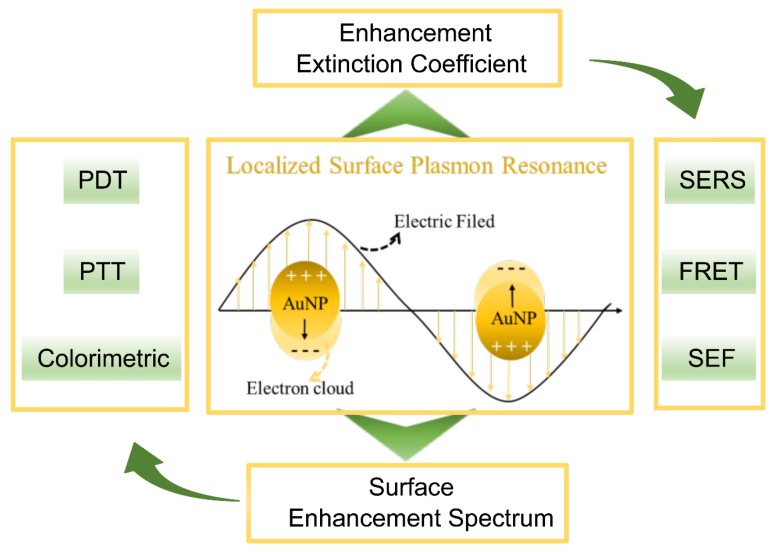
Localized surface plasmon resonance (LSPR) of AuNPs and associated properties.

**Table 1 ijms-21-02480-t001:** AuNPs are used for photodynamic therapy (PDT) and photothermal therapy (PTT) combination.

Gold Nanostructure	Laser (nm)	Coating	Application	Reference
Nanocages	940	Lipid	Hela; PTT/PDT	[51]
Nanocages	980	Lipid	B16F0 melanoma tumors	[51]
Nanocages	790	Hypocrellin and Lipid	Hela; PTT/PDT	[52]
Nanostar	671	Chlorin e6	breast cancer and lung cancer; PTT/PDT	[53]
Nanorods	808	Styrene-alt-maleic acid Indocyanine green	anti-EGFR antibody; PTT/PDT	[54]
Nanorods	810	Rose bengal	Hamster cheek pouches; PTT/PDT	[55]
Nanorods	770	Ce6–pHLIPss Thiol-terminated monomethoxyl	95-C cells; PTT/PDT	[56]
Nanorod	633 and 808	Mesoporous silica Hematoporphyrin	large solid tumors; PTT/PDT	[57]
Nanorods	808+633	Neutrally charged polymers	white outbred male rats with implanted cholangiocarcinoma PC-1; PTT/PDT	[58]
Nanorods	670–710	Sulfonated aluminum Phthalocyanines	human nasopharyngeal carcinoma cells; PTT/PDT	[59]
Nanorod	810 and 670 subsequently	AlPcS4	xenografted mouse tumor; PTT/PDT	[60]
Hollow gold nanospheres	670	pHLIP and Ce6	Hela; PTT/PDT	[61]

**Table 2 ijms-21-02480-t002:** AuNPs are used for clinical trials in tumor therapy and diagnosis.

Name	Composition	Physical& Chemical Property	Application	Phases	Ref.
CYT-6091 (Aurimune)	AuNsphere rhTNF tPEG	Delivery	Melanoma Sarcoma	PhaseⅠcomplete	[140]
Aurolase^®^ therapy	AuNshell Silica PEG	EPR effect Photothermal conversion	Head and Neck Cancer, Lung Tumors, Prostate Cancer	Not Applicable	[141]
PEGylated gold nanoparticles	AuNRod PEG	Photothermal conversion	Deep-tissue Malignancies	Human pilot studies	[142]
NU-0129	Spherical Nucleic Acid Ausphere	Delivery	Glioblastoma Gliosarcoma	Early Phase Ⅰ	[143]

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
