# Peer review of "The Basic Properties of Gold Nanoparticles and their Applications in Tumor Diagnosis and Treatment"

_ijms, 2020, doi:10.3390/ijms21072480_

Round 1
Reviewer 1 Report
The manuscript provides short but informative integration of key properties of gold nanoparticles that are important for their bioapplications, first of all in tumor diagnosis and treatment. It should be noted that several reviews in this field are already published and actively used as sources of information concerning basic physico-chemical properties of AuNPs. See first of all:
The golden age: Gold nanoparticles for biomedicine (DOI: 10.1039/c1cs15237h) – 1,749 citations since 2012
Gold nanoparticles in biomedical applications: recent advances and perspectives (DOI: 10.1039/c1cs15166e) – 1,027 citations since 2012
Gold nanomaterials at work in biomedicine (DOI: 10.1021/acs.chemrev.5b00193) – 511 citations since 2015
Gold nanoparticles for applications in cancer radiotherapy: Mechanisms and recent advancements (DOI: 10.1016/j.addr.2015.12.012) – 202 citations since 2017, etc.
The basic properties of AuNPs is a relatively stable field of knowledge without necessity to revise new data each year. So the indicated above and similar papers will continue to be key sources of information about the AuNPs.
However, these papers consider in detail physical regularities of the unique properties of AuNPs and due to this are not easy for understanding. More simple texts for specialists who works with concrete preparations of GNPs and needs in principal comments about their action are of demand.
The review of Xue Bai et al. accords to this criteria and may be a subject of publication. The authors give shot and clear systematized analysis of the basic properties of AuNPs influencing their diagnostic, theranostic and therapeutic use in oncomedicine. The text is scientifically correct and well structured. However, several revisions would be reasonable before the publication:
1_. Actually the main part of the paper is formed by the references list (18 pages from 29; 299 items). Such high quantity of sources cannot be completely commented in the core text and so cannot be efficiently used by readers. The main recommendation is to reduce this list to 100-150 references. Please exclude the confirmation of one statement by prolonged lists of publications and multiple examples of the use for each preparation. The complete bibliography concerning the AuNPs use in oncomedicine includes thousands of papers, and only a limited part of them may be given in the review. Where it is possible, please address readers to previous reviews summarizing the corresponding issues.
2_. It would be reasonable to give some short list of more detailed publications (reviews) about physical and chemical aspects of AuNPs at the beginning of the review. By this way the interested persons will be re-addressed to corresponding sources of information.
3_. The variety of AuNPs in shape is their common property that is not associated with PDT and PTT use. So the consideration of this variety should be given in the introductory part.
4_. The therapeutic use of AuNPs depends on their own biological activity (toxicity etc.). So it will be useful to integrate the corresponding knowledge in a special part of the review. Actually only approaches to increase the biocompatibility of the AuNPs are listed in the Section 3.1.1, without the estimation of their initial biological properties.
5_. The use of colorimetric response of AuNPs in diagnostics is not always associated with the color change. A lot of assay techniques are based on direct optical detection of AuNPs as labels of the formed complexes. This wide field of R&D should be clearly indicated in the Section 2.1.5 with addressing to key reviews.
6_. The text about selective attachment of 199Au to monoclonal antibodies (lines 273-274) is not clear. Do the authors state that 199Au is more affine to these protein molecules at compared with other atoms? Please check and revise/delete.
7_. Please avoid the demonstration of concrete quantitative parameters of specific studies, as well as these data cannot be transferred to other works. (See consideration of refs. 54, 200, 215, etc.)
8_. The factors causing aggregation of AuNPs and shape of formed complexes are considered in a row of papers. The addressing to some reviews will be better as compared with the actual collection of 29 references about this listed at three lines (184-186).
9_. Some data about clinical trials at Table 2 are based on publications of 2016-2017. Please check the actual status of the indicated preparations.
10_. Please check the references and give complete bibliographic data for all cases.
Author Response
Dear reviewer:
Thank you very much for your careful review and your professional revision suggestions. According to these comments, the manuscript was improved and We submit here the revised manuscript as well as a list of changes.
1_. Actually the main part of the paper is formed by the references list (18 pages from 29; 299 items). Such high quantity of sources cannot be completely commented in the core text and so cannot be efficiently used by readers. The main recommendation is to reduce this list to 100-150 references. Please exclude the confirmation of one statement by prolonged lists of publications and multiple examples of the use for each preparation. The complete bibliography concerning the AuNPs use in oncomedicine includes thousands of papers, and only a limited part of them may be given in the review. Where it is possible, please address readers to previous reviews summarizing the corresponding issues.
According to this very valuable suggestion, we reassessed 299 articles in this paper and excluded the prolonged lists of publications for one statement confirmation and save only one of the most convincing examples for each preparation. What we did was we replaced the specific literature that was listed with some influential reviews and then put the literature in the supplemental material. As a result, we reduced the number of references in this paper from 299 to 146. Please refer to the reference section in revised manuscript.
In the introduction section, we briefly introduce several influential reviews and highlight their salient points. Please refer to line 45-56 in revised manuscript.
2_. It would be reasonable to give some short list of more detailed publications (reviews) about physical and chemical aspects of AuNPs at the beginning of the review. By this way the interested persons will be re-addressed to corresponding sources of information.
From a variety of databases, we searched a large number of review articles on explaining the physical and chemical properties of AuNPs, extracted their topics and presented them in table form. However, due to space constraints, we added them to the supplementary materials. Please refer to supplementary materials.
3_. The variety of AuNPs in shape is their common property that is not associated with PDT and PTT use. So the consideration of this variety should be given in the introductory part.
In the introduction section, we have added a brief introduction on the effect of gold nanoscale shape on its physical and chemical properties and its biomedical applications, including PTT、PDT and mimic enzyme activity. Please tefer to line 56-63 in revised manuscript.
4_. The therapeutic use of AuNPs depends on their own biological activity (toxicity etc.). So it will be useful to integrate the corresponding knowledge in a special part of the review. Actually only approaches to increase the biocompatibility of the AuNPs are listed in the Section 3.1.1, without the estimation of their initial biological properties.
Thanks for your suggestion. We have added the introduction about the biological activity of AuNPs in revised manuscript, please refer to line 400-412. In addition, we re-introduce the biocompatibility of AuNPs in revised manuscript, please refer to line 333-359.
5_. The use of colorimetric response of AuNPs in diagnostics is not always associated with the color change. A lot of assay techniques are based on direct optical detection of AuNPs as labels of the formed complexes. This wide field of R&D should be clearly indicated in the Section 2.1.5 with addressing to key reviews.
First of all, sorry for the misunderstanding caused by our presentation. In the Section 2.1.5, we introduce the application of AuNP-based colorimetric assays in tumor diagnosis and treatment. AuNPs are a perfect candidate for colorimetric assays because of their ultrahigh extinction coefficients. In general, substrates with higher extinction coefficients can provide higher detection sensitivity. AuNPs with different particle size have different extinction coefficients, and different extinction coefficients make the AuNPs show different colors under the same incident light. So the AuNP-based colorimetric assays in diagnostics is related to the color change (Chem. Rev. 2015, 115, 10575−10636).
As for other detection methods that directly apply the optical properties of AuNPs, tare respectively introduced in the corresponding optical properties, such as AuNPs-based fluorescent assays are introduced in the Section 2.1.2; AuNPs-based SERS assays are introduced in the Section 2.1.1.
6_. The text about selective attachment of 199Au to monoclonal antibodies (lines 273-274) is not clear. Do the authors state that 199Au is more affine to these protein molecules at compared with other atoms? Please check and revise/delete.
First of all, we apologize for the confusion caused by the unclear expression in the text. The reason that 199Au are better for SPECT imaging than 198Au is that they can emits β-particle with a maximum energy of 0.45 MeV and a 208 keV γ-ray and 198Au can emits β-particle with a maximum energy of 0.96 MeV and a 412 keV γ-ray, not because they are more prone to antibody binding. In fact, the original results demonstrate that coupling tumor-specific antibodies to gold nanoparticles increases their targeting, but do not discuss which gold atoms the antibody is attached to.In order to express more clearly that the reason why 199Au are more suitable for SPECT imaging is because of the β-particle energy they emit, rather than because they are more easily bound to tumor antibodies, we removed the statement that AuNPa containing 199Au is bound to antibodies. Please refer to line278-284 in revised manuscript.
7_. Please avoid the demonstration of concrete quantitative parameters of specific studies, as well as these data cannot be transferred to other works. (See consideration of refs. 54, 200, 215, etc.)
We have examined the concrete quantitative parameters from the references cited in this paper and have hidden the specific data which is not very necessary. Please refer to line262-263,272-273,308-311 and 126-129 in revised manuscript.
8_. The factors causing aggregation of AuNPs and shape of formed complexes are considered in a row of papers. The addressing to some reviews will be better as compared with the actual collection of 29 references about this listed at three lines (184-186).
Thank you very much for your professional advice,We have replaced the references listed in this paper with representative reviews. Please refer to line 198-201 in revised manuscript.
9_. Some data about clinical trials at Table 2 are based on publications of 2016-2017. Please check the actual status of the indicated preparations.
Thank you very much for your important Suggestions. We have researched the application of AuNPs in clinical and clinical trials based on http://clinicaltrials.gov/ and updated the table in the paper. Please refer to the first and second rows of the table 2.
10_. Please check the references and give complete bibliographic data for all cases.
According to the Suggestions, we carefully rearranged and revised the number and format of references. Please refer to the references section.
Reviewer 2 Report
The review of Xue Bai et al. is very interesting since it is focused on a very interesting issue. My advice is to publish the review after a minor revision.
Minor revisions:
In the abstract, line 26: please correct with “physicochemical”
Page 3 line 66: please correct with “which is called”
Page 3 lines 92, 93: Since scattering may occur both for particle smaller (Rayleigh scattering) and bigger (Mie scattering) than the wavelenght of incident light, please be careful with the statement in the sentence.
Page 4 lines 100-102: please check the grammar
Page 4 line 144: please correct with “This method is also used”
Page 5 lines 147-163: please revise the paragraphs. Delate identical sentences and check for the right references.
Page 5 lines 174, 175: please check the sentence accuracy
Page 6 line 202: please correct with “As a photosensitive”
Page 8 line 247: 198 as superscript
Page 8 line 282: Auger with capital letter
Page 10 line 337: please correct with “through”
Page 11 lines 373, 374: which are the tumor therapy strategies already used in clinical application (routinely)? If there are, do not limit table 2 to AuNPs used in clinical trials. Otherwise, just state that they are used in clinical trials
In the whole text: where applicable, please correct co-works with co-workers.
Author Response
Dear reviewer:
Thank you very much for your professional revision suggestion. According to this comment, the manuscript was improved and we submit here the revised manuscript as well as a list of changes.
In the abstract, line 26: please correct with “physicochemical”
We have corrected it, please refer to line 26 in revised manuscript.
Page 3 line 66: please correct with “which is called”
We have corrected it, please refer to line 87 in revised manuscript.
Page 3 lines 92, 93: Since scattering may occur both for particle smaller (Rayleigh scattering) and bigger (Mie scattering) than the wavelenght of incident light, please be careful with the statement in the sentence.
Thank you very much for asking such a rigorous question. We have emphasized that the scattering occurs for particle smaller than the wavelenght of incident light is Rayleigh scattering. Please refer to line 112 in revised manuscript.
Page 4 lines 100-102: please check the grammar
Page 4 line 144: please correct with “This method is also used”
We have checked the grammar and correct both of them, please refer to line 199-122 and 156 in revised manuscript.
Page 5 lines 147-163: please revise the paragraphs. Delate identical sentences and check for the right references.
We have corrected it, please refer to line 164-177 in revised manuscript.
Page 5 lines 174, 175: please check the sentence accuracy
We have corrected it, please refer to line 183-189 in revised manuscript.
Page 6 line 202: please correct with “As a photosensitive”
We have corrected it, please refer to line 216 in revised manuscript.
Page 8 line 247: 198 as superscript
We have corrected it, please refer to line 261 in revised manuscript.
Page 8 line 282: Auger with capital letter
We have corrected it, please refer to line 296 in revised manuscript.
Page 10 line 337: please correct with “through”
We have corrected it, please refer to line 366 in revised manuscript.
Page 11 lines 373, 374: which are the tumor therapy strategies already used in clinical application (routinely)? If there are, do not limit table 2 to AuNPs used in clinical trials. Otherwise, just state that they are used in clinical trials
According to the FDA's website, there is currently no AuNPs-based drug that is formally used in clinical oncology. We have updated the status of ongoing clinical trials based on data from the FDA website. Please refer to the first and second rows of the table 2.
In the whole text: where applicable, please correct co-works with co-workers.
We have correct them, please refer to line 126,129 and131 in revised manuscript.
Round 2
Reviewer 1 Report
The review has been substantially revised by the authors taking into account the recommendations made. Its current version may be recommended for publication.